# Feature Channel Expansion and Background Suppression as the Enhancement for Infrared Pedestrian Detection

**DOI:** 10.3390/s20185128

**Published:** 2020-09-09

**Authors:** Shengzhe Wang, Bo Wang, Shifeng Wang, Yifeng Tang

**Affiliations:** 1National Demonstration Center for Experimental Optoelectronic Engineering Education, School of Optoelectronic Engineering, Changchun University of Science and Technology, Changchun 130022, China; sz.wang@mails.cust.edu.cn (S.W.); b.wang@mails.cust.edu.cn (B.W.); tangyf@mails.cust.edu.cn (Y.T.); 2Key Laboratory of Optoelectronic Measurement and Optical Information Transmission Technology of Ministry of Education, School of Optoelectronic Engineering, Changchun University of Science and Technology, Changchun 130022, China

**Keywords:** feature fusion, background suppression, pedestrian detection, infrared image enhancement, fusion of saliency maps

## Abstract

Pedestrian detection is an important task in many intelligent systems, particularly driver assistance systems. Recent studies on pedestrian detection in infrared (IR) imagery have employed data-driven approaches. However, two problems in deep learning-based detection are the implicit performance and time-consuming training. In this paper, a novel channel expansion technique based on feature fusion is proposed to enhance the IR imagery and accelerate the training process. Besides, a novel background suppression method is proposed to stimulate the attention principle of human vision and shrink the region of detection. A precise fusion algorithm is designed to combine the information from different visual saliency maps in order to reduce the effect of truncation and miss detection. Four different experiments are performed from various perspectives in order to gauge the efficiency of our approach. The experimental results show that the Mean Average Precisions (mAPs) of four different datasets have been increased by 5.22% on average. The results prove that background suppression and suitable feature expansion will accelerate the training process and enhance the performance of IR image-based deep learning models.

## 1. Introduction

Pedestrian detection is a vital research topic in the field of computer vision, one of significant theoretical interest, and with various applications. Different types of sensing devices can be used to capture information from diverse dimensions. Of these, the specialized use of infrared imagery for pedestrian detection in images is of particular interest, since they contain different information from the visible light images. This information, especially those from thermal radiation, is available to be employed to detect targets with higher temperatures. The infrared radiation of some wave band can penetrate through the cloud and mist. Moreover, information from infrared imagery can be employed both day and night. Therefore, such types of detection can be used not only in intelligent surveillance systems, but also in the advanced driver assistance systems (ADAS) for greater safety for humans and vehicles.

In the infrared (IR) spectrum approach, the IR cameras receive radiation emitted from the scene in the daytime and nighttime conditions [1,2], and their intensity distribution is not sensitive to the illumination of the scene or color of the object [3,4,5]. Human candidate pixels are supposed to be with higher intensity in the IR images because the temperature of the human body is usually higher than that of the background [6]. Although an increasing number of theories and methods have been put forward as solutions for visible light classification and detection problems [7,8,9,10], those for IR imagery detection have never been proposed in a systematical manner. In general, the IR spectrum can be classified into four sub-bands, such as near-IR, short-wave IR, medium-wave IR, far-infrared [11,12]. Among these bands, humans are more physically visible in the far-infrared camera than in the other cameras [12,13]. However, there are two main factors that have hindered the development of detection based on the far-infrared imagery. First, the performance of the far-infrared pedestrian detection could be influenced by low image resolution, low contrast, and the large noises of far-infrared images. Besides, the performance of the detection is also affected by the high temperature of the cluttered background caused by hot weather in the day time and radiation from other sources like the sun or the lamps [13].

Current research on human detection can be broadly classified into two categories: Expert-driven and data-driven learning approaches [14].

In the expert-driven learning approaches, feature extraction and classification are two necessary steps. Feature extraction of the human body in the image is carefully designed in order to transform the raw image into a feature map, which can be recognized by the classifiers. The algorithm of pattern extraction is designed or adjusted by experts, according to the discipline of image processing or the features of the object itself. Many efforts have been made to develop shape features that are not sensitive to brightness, orientation, or scale. However, the generalization capacity of expert-driven learning approaches is often very limited, and some of them are designed to solve specific problems [15]. For instance, Bassem et al. distinguished pedestrians from other objects by utilizing a SURF-based feature based on the assumption that a pedestrian’s head appears as a light region [16]. However, the shape information of a pedestrian’s body is not considered, leading to unreliable classification in complex cases with noises or pedestrians with headgear [16]. Kwak et al. designed an image feature based on the prior knowledge that pedestrian form is characterized by higher intensity relative to the background [17]. However, this approach fails to produce accurate classification results when a pedestrian’s motion is large [17].

The data-driven learning approach, especially deep learning, has attracted a great deal of interest in recent years, not only for its outstanding performance in both data classification and object detection, but also for its ability to learn features and patterns from raw images without human support. Convolutional neural networks (CNNs) were first designed according to the principles of information processing and communication in a biological nervous system [18]. AlexNet, the winner of the ImageNet Large Scale Visual Recognition Challenge in 2012, was generated by several techniques on a CNN to enhance its abilities. Visual Geometry Group at the University of Oxford designed VGGNet by increasing the depth of the CNN and using smaller kernel size [19], but the computational cost of a very deep CNN is still too expensive for real-time applications. He et al. created ResNet by adding a residual learning block, which made the ResNet more effective [20]. All of these CNN-based approaches offered increased performance on classification problems—but to solve detection problems, they required sliding window detection to scan all the images, which incurred a prohibitive computational cost. As a CNN’s depth grows and its architecture becomes more complex, the computational cost increases significantly. In addition, a data-driven learning approach tends to require vast amounts of data, which makes the development of a product much more complicated. Furthermore, the region proposal methods are not suitable for IR images, since IR imagery is based on intensity distribution with only one channel, which many different characteristics from the visible imagery. Future work will focus on the characteristics of IR imagery and the methods for accelerating the training process. 

Therefore, in this research, an assumption has been given that appropriate expert-driven features can help with the extraction of CNN features and accelerate the training process of the model. In this work, an IR image channel expansion enhancement approach is proposed to address the problems described above. The domain expertise of the expert-driven approach is applied to pre-extract the features of the IR imagery, thereby speeding up the training process of the data-driven detection training process. At the same time, we employed several visual saliency methods to shrink the area that remains to be searched, in order to simulate how the human eye pays attention to an object. Current visual saliency methods cannot handle the many types of IR images, because the IR sensors are all different. This issue is mitigated in this study, and a fusion method is proposed to solve this problem. Our research is novel compared to previous works in the following four ways:This is the first research on IR pedestrian detection using an IR image enhancement unit to extract the artificial features as the input of the CNN, and using a feature optimization network to output a further optimized feature map.In this research, experiments were carefully designed, and they proved the assumption that appropriate expert-driven features can help with the extraction of CNN features and accelerate the training process of the model.The detection performance has been improved by the model proposed in this research, compared with the baseline methods and original region proposal networks, which was proved by the experiments.A new saliency fusion method was designed in this research to suppress the background. This fusion method proved that background suppression could improve the performance of pedestrian detection by reducing the negatives, according to the experiments.

The remainder of this paper is organized into four sections. Section 2 briefly reviews the existing work related to this study. In Section 3, the details of our method are described. Section 4 presents the experimental results with findings and analysis. Section 5 offers our conclusions.

## 2. Related Work

This study focuses on enhancing through channel expansion and saliency methods the performance of CNN-detection on IR imagery. Therefore, we briefly review the related work in terms of features in expert-driven approaches and visual saliency methods.

### 2.1. Features in Expert-Driven Approaches

Many researchers have put great effort into developing the extraction of shape features, as the shape of a human is a critical feature for human classification, particularly in infrared images. 

The histogram of oriented gradients (HOG) and its variants are among the most frequently used techniques employed in machine learning. The usage of HOG did not become widespread until 2005, when Dalel and Triggs put forth the importance of the gradient feature in accurately representing the appearance and shape of an object. Therefore, HOG was first designed for classifiers, for example, support vector machine (SVM), to recognize the class of target [21]. However, this feature algorithm is not sensitive to slight movements, illumination, or color of pedestrians, since its range of sampling in the spatial domain is relatively large. Frederic et al. used the HOG feature descriptor on IR image pedestrian detection [22]. However, this descriptor is comparatively sensitive to noises and the truncation of the object. Meanwhile, the additional computation cost of this descriptor leads to poor performance in real-time applications.

Local binary pattern (LBP) is a local textural feature with substantial rotation invariability and gray-level invariability. It was first put forth by Ojala, Pietikäinen, and Harwood in 1994 in order to extract texture in small parts of an image. Ojala et al., employed a circle mask of operator for sampling in order to deal with various sizes and frequencies [23]. To handle a large number of binary patterns, they put forth a uniform pattern to reduce data size. In recent years, different kinds of variants have been developed to solve particular problems. The center symmetric local binary pattern (CSLBP) is designed by comparing the center-symmetric pairs of pixels with a central pixel rather than comparing each pixel with the center [9]. The CSLBP approach maintains certain characteristics of rotation invariability and gray-level invariability, while reducing computational cost [7,24]. In multi-scale block local binary pattern (MB-LBP), developed by Liao et al. [25], the computation is performed based on average values of block sub-regions, instead of individual pixels [25]. The MB-LBP operator provides a more complete image representation than the original LBP operator, since it can extract not only microstructures, but also macrostructures of image patterns. Furthermore, MB-LBP can be computed very efficiently with integral images, which makes it highly suitable for real-time conditions, in particular, pedestrian detection.

Over the last two decades, many other interesting approaches have been proposed for detection in various applications. These include the edgelet features proposed by Wu and Nevatia [26], the scale-invariant orientation features designed by Mikolajczyk et al. [27], the Haar wavelet developed by Mohan et al. [28], and the motion-enhanced Haar features modified by Viola [29].

Because the aforementioned approaches are based on an empirical model, they may change significantly with the season, weather, and background, especially when it comes to IR imagery conditions.

### 2.2. Visual Saliency Methods

In recent years, visual saliency methods have been developed to extract the region of interest. This kind of approach, takes raw imagery as input and calculates the saliency map of the image via stimulating the visual attention mechanism—which means that the model evaluates each pixel and generates a score for it. This score is usually in proportion to the possibility of the pixel being a part of the object.

One of the earliest saliency models, proposed by Itti et al., is an implementation of earlier general computational frameworks and psychological theories of bottom-up attention-based on center-surround mechanisms [30]. In the model, salient object detection is defined as capturing the uniqueness, distinctiveness, or rarity of a scene. In this model, center-surround differences and normalization are executed on intensity, orientations, and colors, in order to obtain feature maps. The feature maps are transformed into conspicuity maps via across-scale combinations and normalization, and then combined into a saliency map [30]. Itti et al.’s saliency model can be calculated in a short period of time, and is not sensitive to noise.

Another bottom-up visual saliency model, Graph-Based Visual Saliency (GBVS), was proposed by Harel, Koch, and Perona [31]. It consists of two steps: Forming activation maps on certain feature channels, and then normalizing them in a way that highlights conspicuity and allows combination with other maps [2]. This model is simple, and biologically reasonable as the simulation of human visual attention.

A technique to highlight sparse salient regions was proposed by Hou et al., which is referred to as the image signature [32]. This model approximates the foreground of an image within the theoretical framework of sparse signal mixing.

Previous studies show that visual saliency methods are suitable for grayscale images, especially IR images, without color distribution [33]. Moreover, there were some recent works focusing on object detection with saliency methods. Sheng Xing and Wu Ying proposed a salient object detection method via low-rank matrix recovery, which could achieve comparable performance to the previous methods without help from high-level knowledge [34]. Qi Zheng, Shujian Yu, and Xinge You considered the interrelationship among elements within the sparse components, and used a coarse-to-fine framework to achieve enhanced performance [35]. Li Junxia and Yang jian et al., consider the saliency fusion as a subspace decomposition problem and proposed a novel sparse and double low-rank model to combine various saliency maps [36]. Dingwen Zhang and Junwei Han et al., developed a framework to learn deep salient object detectors without requiring any human annotation [37].

### 2.3. Regions with CNN Features Method

The specialization of detection networks to classification networks begins with the region proposal method. The selective search was first put forth to provide several candidates for classifiers to recognize, by cutting the image into different areas with the same color, texture, and gradient features [8,38]. In 2014, Girshick et al. employed AlexNet to generate features, utilized selective search to propose candidates, and used SVM to classify the candidates, which is integrated as a region-proposal CNN (R-CNN) [10]. In 2015, a Fast R-CNN was proposed with two output vectors—softmax probabilities and per-class bounding-box regression offsets—which creates an end-to-end architecture with a multi-task loss [39]. He’s team used region proposal networks to generate candidates and a CNN to the extract feature map, thereby making the R-CNN even faster in real-time detection problems [40].

The present study combines expert-driven and data-driven approaches to provide an economical, effective solution for IR pedestrian detection. Gradient features and MBLBP operator are used to generalize the IR data from different sensors, while expanding the channels of the images. Visual saliency models are used to restrain the background and reduce the false positives by shrinking the size of the candidate regions. Fast R-CNN and Faster R-CNN detection models are employed to verify the rationality of our model, and through experiments, demonstrate its accuracy and its economy in terms of computing resources.

## 3. Background Suppression and Channel Expansion Enhancement Approach

In this section, an IR imagery enhancement approach based on feature expansion and background suppression for infrared pedestrian detection is presented. Figure 1 is an introduction to this approach.

The proposed approach consists of two main procedures: The feature channel expansion and background suppression. The input of this system is raw IR imagery of scenes containing pedestrians and background. The system generates as output both the position and probability of the pedestrians predicted in the images.

Because the size of the input image varies depending on the type of sensor, the input image is first normalized on size and transformed into a fixed size (height of 1000 pixels and width of 1600 pixels). This size normalization is used to unify the sizes of images from different datasets. Moreover, there will be different performance if we apply the same convolution kernel or filtering kernel on images with different sizes. Therefore, size normalization can improve the robustness of the system. Feature expansion is designed to generate artificial feature maps. The model expands the channels of IR images with feature maps, because this research makes the assumption that appropriate expert-driven features can help with the extraction of the CNN features and accelerate the training process of the model. The selected features will be introduced in Section 3.1. The background suppression uses the original infrared images to estimate the foreground and background, and transforms those regions into a saliency map to generate the enhanced images. The background suppression can remove most of the cluttered background, therefore, improve the precision. A fusion algorithm for saliency maps will be introduced in Section 3.2. This system could accelerate the training process of the region proposal networks (RPNs), as well as improve the final performance.

As is shown in Figure 1b, the proposed model contains three parts: IR image enhancement, feature optimization network, region proposal networks. Feature optimization network takes artificial feature maps as input and output a further optimized feature map. This part has a structure similar to VGGNet, with the architecture parameters, as shown in Table 1. Feature optimization network is vital, since the artificial features are not perfect enough for classification. The region proposal network employs two convolution layers to generate the object map with 18 channels, uses a softmax layer to pick up proposal regions, and uses a regression branches to adjust the positions and sizes of the proposal regions [40]. Finally, two full connection layers with 4096 units, one with 21 units and one with 84 units, are used to output the bounding boxes and classes.

### 3.1. Feature Channel Expansion

It is assumed that appropriate expert-driven features can help with the extraction of the CNN features and accelerate the training process of the model. In order to verify the assumption, two feature descriptors are employed to extract information from the IR image. Expert-driven approaches often use shape information (which represents overall features of the objects) and texture information (which represents the local features of the objects) [41]. In our work, gradient magnitude is selected to preliminarily extract the shape information of the object, because it is proved that gradients contain the discriminative information to reflect the target variations and distinguish the target from background clutter [42]. Furthermore, other shape feature operators, such as HOG, are statistics and variants of the gradients which contain gradients information, but they bring more calculation cost [22]. The multiscale block LBP (MB-LBP) of the input image is also computed because of its high efficiency in both texture feature extraction and computation [25]. 

The algorithm of feature channels expansion is shown in Figure 2 and Figure 3. Bilinear interpolation is first used to normalize the size of the image. A 3 × 3 Gaussian filter is employed to suppress the noise in the original images. Then histogram equalization is performed to improve the contrast of the IR images. Therefore, the gradient feature is enhanced and can be extracted effectively [43]. The performance of the gradient extraction can be affected by serious noise, which is suppressed by an ideal lowpass filter and a Laplacian filter. Then an intensity local normalization is carried out on the gradient map, in order to minimize the effect of the intensity-level changes [41]. MB-LBP feature map is calculated, according to the method introduced in Section 3.1.2. The artificial feature maps are generated by arranging these two maps together with the original intensity distribution, thereby serving as the three channels of the output image.

#### 3.1.1. Image Gradient Channel Computation

The image gradient is usually defined as the directional change in the gray level. Suppose that $G_{x}$ and $G_{y}$ are the horizontal and vertical gradients of the input image I(x,y). The image gradients can then be calculated by the following Equations.
(1)$$\nabla f(x,y)={\left[G_{x},G_{y}\right]}^{T}={\left[\frac{\partial f}{\partial x},\frac{\partial f}{\partial y}\right]}^{T}$$

In the digital image, the differential can be replaced by difference.
(2)$$G_{x}(x,y)=f(x,y)-f(x-1,y)$$
(3)$$G_{y}(x,y)=f(x,y)-f(x,y-1)$$

The gradient magnitude can be computed as:(4)$$G(x,y)=\sqrt{G_{x}{\left(x,y\right)}^{2}+G_{y}{\left(x,y\right)}^{2}}$$

The intensity normalization of the gradient can be described as:(5)$$G_{0}(x,y)=255\times \frac{G(x,y)-\min\{G(x,y)\}}{\max\{G(x,y)\}-\min\{G(x,y)\}}$$

#### 3.1.2. Texture Channel of MB-LBP Computation

The original LBP operator labels the pixels of an image by thresholding the 3×3 neighborhood of each pixel with the center value and considering the result as a binary string or a decimal number [23]. Multi-scale LBP is an extension to the basic LBP, with respect to neighborhoods of different sizes [24].

Figure 4 shows an example of eight neighborhood pixels. Suppose that $I_{c}$ is the intensity value of the center pixel and that $I_{p}$ is the intensity value of the neighborhood pixels. The binary sequence can be generated as:(6)$$T=s(I_{1}-I_{c}),s(I_{2}-I_{c}),\ldots ,s(I_{8}-I_{c})$$
(7)$$s(x)=\left\{\begin{array}{l} 1,x\geq 0\\ 0,x<0 \end{array}\right.$$

The binary sequence can be transformed into a unique LBP number that represents the local image texture [23], which can be described by Figure 4:(8)$$LBP=\sum_{p=1}^{8}s(I_{p}-I_{c})\times 2^{p-1}.$$

In the MB-LBP, the comparison is between average gray-values of sub-regions, instead of between single pixels in one sub-region. As shown in Figure 5, the entire filter consists of nine blocks [23]. Because the range of the MB-LBP operator can be adjusted, it can not only represent the local feature of the region, but also the macrostructure of the images. Therefore, this algorithm is more likely to extract the pedestrian features. 

### 3.2. Background Suppression

The background suppression component of our system combines two types of saliency algorithms to suppress the background. These two saliency algorithms generate the GBVS map and Itti-Koch map to estimate the possibility that a particular pixel or region belongs to an object. Since human beings are distinct in IR images, due to their high intensity, saliency approaches are more suitable for IR imagery.

Algorithm 1 illustrates the proposed background suppression according to GBVS and Itti-Koch Map. These two maps are first normalized, because the range of the two maps is different. Next, we compare the two maps in each sub-region, and pick the larger one of the simple sub-region as the saliency value. The intensity distribution of the original image is then modulated by the value, and the background suppressed image is generated.

As shown in Figure 6, the GBVS approach tends to capture a single connected domain, which means that false positives and false negatives are more likely to occur, leading to a decline in recall value. The Itti-Koch saliency approach tends to capture multiple regions, but it may capture other classes of objects, such as cars, trees, or even buildings, which reflect the sunlight. The fusion algorithm is carefully developed in this work, because we clearly want to avoid the truncation of the pedestrian.

**Algorithm 1** Background Suppression according to GBVS and Itti-Koch Map**Input:** an original infrared image *I*_0_**Output:** a background-suppressed image1: the standardized image *I* ← standardize Image (*I*_0_)  //Both histogram equalization (HE) and size normalization  //*Step 1:* GBVS map and Itti-Koch map generation:2: GBVS map ← GBVS(*I*)3: Itti-Koch map ← Itti-Koch(*I*)4: GBVS map & Itti-Koch map normalization  //*Step 2:* GBVS map and Itti-Koch map fusion:5: **for each** unknown pixel **do**6: **for each** map **do**7: calculate average saliency in 5 × 5 neighborhood8: **end for**9: find the largest average saliency as the fusion saliency value10: new image ← *k* × original image × saliency value11: **end for**12: generate all pixels in the new image13: **return** the mapping of the new image

## 4. Experiments

We performed four experiments for the IR image channel expansion and background suppression of pedestrian detection tasks. These four experiments, including visual comparison, accuracy evaluation, comprehensive evaluation, and the requirement for data, were designed to evaluate the efficiency of our approach from several aspects. First, we will describe the databases and the environment employed in this work, as well as the quantitative evaluation methods used.

### 4.1. Introduction of Experimental Datasets and Environment

Four databases were used for the performance evaluation in order to estimate the generalization capability of our model. The experiments were carried out on four widely-used benchmark datasets: the LSI far infrared pedestrian dataset (LSI), South China University of Technology far infrared ray pedestrian dataset (SCUT), CVC14 day-night pedestrian sequence dataset (CVC14), and Changchun University of Science and Technology far infrared ray (CUSTFIR) pedestrian dataset.

The LSI dataset was obtained from a vehicle driven in outdoor urban scenarios. Images were acquired with an Indigo Omega Imager, with a resolution of 164 × 129 pixels. The training set contained 10,208 positives and 43,390 negatives, while the test set contained 5944 positives and 22,050 negatives [44,45].

SCUT is a large far-infrared pedestrian detection dataset [46]. It consists of 11 h-long image sequences at a rate of 25 Hz generated by driving through diverse traffic scenarios at under 80 km/h. The image sequences were collected from 11 road sections in four kinds of scenes: Downtown, suburb, expressway, and campus. There were 211,011 frames annotated manually, for a total of 477,907 bounding boxes around 7659 unique pedestrians.

The CVC14 day-night pedestrian sequence dataset was obtained on a moving platform on the street in both daytime and nighttime settings [47]. Overall, 7085 images were recorded with approximately 1500 pedestrians annotated manually for each sequence. In this experiment, we used two typical sequences.

CUSTFIR is a dataset obtained by our team with an automotive IR camera developed by iRay Technology, in Shanghai, China. The IR camera has a resolution of 352 × 288 and a sampling frequency of 25 Hz. This dataset contains both daytime and nighttime sequences. There are 2836 images with 6408 pedestrians included. All images are recorded with a one out of 10 sampling. The ground truths of this dataset are annotated manually by four different volunteers. The dataset contains two different scenes with irregular background, such as trees, cars, buildings, and sky. The two scenes are on different roads of Changchun University of Science and Technology. The datasets contain sequences of both winter and summer, and it contains sequences of both day and night. Some examples of CUSTFIR pedestrian dataset are shown in Figure 7, where the red boxes are the ground truth.

The ground truths of each pedestrian are the average of annotations given by the four volunteers. When the ground truths are manually annotated, we have defined a threshold for the minimum height of the pedestrians. That is, when we did the experiments, we ignored the ground truths whose bounding box’s height is smaller than 30 pixels.

All scenarios of the four datasets are realistic scenarios. The basic information on the infrared pedestrian datasets used in the experiments is shown in the Table 2.

Each experiment consisted of image enhancement and deep learning with RPNs. The former was conducted on the MATLAB platform on a desktop PC with a 3.1 GHz quad-core Intel Core i5 processor and 8 GB RAM. The latter was conducted on the Torch platform on a professional image-processing PC with a GeForceRTX2080Ti and 11 GB RAM.

### 4.2. Introduction of Quantitative Evaluation

This paper uses quantitative evaluation methods for a classical detection task to gauge the performance of our approach. According to Pascal criteria, the detection results can be classified into four situations: True positives (TP), true negatives (TN), false positives (FP), and false negatives (FN). For a single class detection task, these four situations can be defined as follows:A true negative is a case where the background area was recognized as a background region.A true positive is a case where the human area was correctly recognized as a human region.A false negative is a case where the human area was recognized as a background region.A false positive is a case where the background area was recognized as a human region.

The relationship among four situations can be represented by the Venn diagram in Figure 7 and the chart in Table 3.

In addition, based on the aforementioned TP, TN, FP, and FN, we used the following four criteria to measure accuracy [6]:(9)$$Precision=\frac{\#TP}{\#TP+\#FP}$$
(10)$$Recall=\frac{\#TP}{\#TP+\#FN}$$
(11)$$F-measure=2\times \frac{Precision\times Recall}{Precision+Recall}$$
where #TP, #TN, #FP, and #FN indicate the numbers of TP, TN, FP, and FN, respectively. In the detection task in our work, the #TN is uncountable, because there are no labels for the background samples; therefore, the Accuracy is estimated as the average of best Precision and best Recall. The minimum and maximum values of Precision, Recall, and F-measure are 0 (%) and 100 (%), where 0 (%) and 100 (%) represent the lowest and highest accuracies. 

The Precision–Recall Curves (PR Curves) are based on the aforementioned Precision and Recall. In the final step of the detection, 11 thresholds are set to generate models of different sensitivity. If the threshold was set too high, there would be more FNs and less TPs. If the threshold was set too low, there would be more FPs and less TNs. Therefore, a value, referred to as Mean Average Precision (mAP), should be calculated for each model.
(12)$$\mathrm{mAP}=\int_{0}^{1}P(R)\;dR$$
where R indicates the recall drawing in the PR Curves, and P indicates the precision in the PR Curves.

### 4.3. Visual Comparison

In this experiment, images were presented at different steps of the process, in order to show the visual results in an intuitive way. Figure 8, Figure 9, Figure 10, Figure 11 and Figure 12 show the results by using the LSI datasets. Figure 13 shows the set of typical visual results from the process of the CVC-19 benchmark.

Figure 8 shows a normal example of the model performance of both background suppression and channel expansion. When it comes to a suitable circumstance with a high contrast ratio, the area of vision is reduced rapidly; only the pedestrian is preserved in the image. It can be observed that the image of another radiation source (the ceiling lamp), the structure of the background, and the shadow (reflected radiation) of the pedestrian are properly filtered by the background suppression. The feature algorithms can extract the information of both the edge and texture. Figure 9 shows the model performance with the situation of two targets in two separate domains. It can be observed that both of the pedestrian areas are preserved, with the middle part of the image between two target domains being filtered. Figure 8 and Figure 9 are both taken in indoor conditions, which provide higher contrast and intensity difference between the background and the target.

Figure 10 shows the model performance in the situation of an outdoor image with lower contrast and the implicit difference between the pedestrian and the background. It can be seen that the ground is nearly the same grayscale as the pedestrian with high reflected radiation from the sunshine. In this case, the background suppression approach filters most of the area of the sky and restricts half of the area of the ground, with the pedestrian being preserved. The MB-LBP channel performed the best of the three feature algorithms, especially when the magnitude of gradient was not so obvious.

Figure 11 shows the model performance at the edge of the image, when ground truth multiple targets appear in the same image. In this case, there are three pedestrians, of which one is truncated by the edge. This caused the worst performance of the single saliency methods. In our work, however, the pedestrian on the right is just a bit weakened, instead of being removed. This is because our combination algorithm uses the saliency score as the power to shrink the area of interest. It is designed carefully to protect some area, which is estimated as the background, but is still close to the boundaries of the foreground.

Figure 12 shows an outdoor situation on a sunny day. It can be seen that the background near the pedestrian is complex, with different objects overlapping. The difference between this figure and Figure 10 is that the pedestrian is embedded in the middle of the tough background, which is nearly the same grayscale as the pedestrian. In this situation, only a limited area of background is restricted. The target is easier for detection in the feature maps, due to the LSB feature and magnitude of gradient maps, which will be quantitatively proved in Section 4.4 in a rigorous way. 

### 4.4. Quantitative comparison of Precision, Recall, Accuracy, and F-Measure

The next experiment used the LSIFIR dataset, CVC-14 database, SCUT database, and CUSTFIR dataset to quantitatively estimate the performance difference between the model with channel expansion and background suppression (our work) and the model with the only Faster R-CNN. The model with only the channel expansion approach was also tested in order to show the effect of background suppression. For the LSIFIR dataset, we ran our approach and Faster R-CNN five times to obtain the mean and standard deviation. The Precision, Recall, Accuracy, and F-measure among the four datasets are shown in Table 4.

Table 4 indicates that the proposed approach outperforms the other approaches, which demonstrates that the proposed enhancement model works more efficiently than the original image on the task of infrared pedestrian detection. Moreover, the average performance of the proposed approach was compared with the model with only Faster R-CNN and two other baselines. Two involved approaches were run five times with the LSIFIR dataset. In addition, the model with only channel expansion was tested to show the independent performance of two proposed enhancement approaches. The data show that the channel expansion method improved the model performance in precision, while decreasing the performance of recall. This means that the model is more skeptical and cautious, but is more likely to miss pedestrians at the best point of the PR Curve. The F-measures increase in model with only feature expansion of all datasets—this indicates that the better performance of the model, and proved the assumption that appropriate expert-driven features can help with the extraction of the CNN features and accelerate the training process of the model. The background suppression method improved the performance of recall by shrinking the range of detection, which made the detection more efficient. This is because if the range of background is shrinking, there will be fewer negatives in the detection. Regarding other aspects, according to the calculation of the recall in Equation (10), the background suppression improves the recall by reducing the #FN, since more negatives are suppressed. The synthetic metrics of F-measure showed that the proposed models performed better than the original model overall, since F-measure is the comprehension and the reciprocal average of the precision and recall, according to the Equation (11).

### 4.5. Comprehensive Comparison of mAPs

The mAPs were calculated from the area under the PR Curve in order to evaluate the general performance of all the approaches, which is showed in Table 5. Three models were tested with four datasets.

Figure 14, Figure 15, Figure 16 and Figure 17 exhibit the PR Curve with precisions as the abscissa against recalls as the ordinate on LSIFIR, CVC-14, SCUT, CUSTFIR datasets. Each figure contains three curves of the proposed approach, the model with channel expansion only, the model with Faster R-CNN only, and other baselines like ResNet and VGGNet. It can be observed that the performance of the approaches converges in the low precision area, but tends to be different in the optimal area and the low recall area. In general, the average performance of the proposed approach is better than that of the original model, since the area under the PR Curve (AUC) of our approach is larger than that of the original model. This is because the models of our research provide more pedestrians detected, when keeping the accuracy and the same possibility of making mistakes. This proved the assumption that appropriate expert-driven approaches can help with the extraction of CNN features and accelerate the training process of the model.

### 4.6. The Requirement for the Data Test

In this experiment, mAP is considered as the function of the amount of training data. Two curves are drawn in order to show the difference in data requirements among the approaches, which takes the mAP as the ordinate against the epoch as the abscissa.

Figure 18 shows the mAP against the epoch curve by three models on two typical datasets. In order to show the progress in the model training time, it’s defined in this research that the time for comparison is the time when the curve reaches 90% of the final average mAP value. It is clear that the training of models with channel expansion is one epoch faster than that of the original one, according to the guidelines in purple. In addition, the curves of our approach flatten sooner than the curves of the original models. This demonstrated that our approach could accelerate the training process without additional data. It can also be seen that the curves of our approach tend to decrease at the end of the training, instead of maintaining their level. This may be because the model of our approach is more likely to be overfitted with excessive data.

## 5. Conclusions

In this research, an enhancement method for pedestrian detection with channel expansion and background suppression approaches has been studied in an IR imagery environment. The magnitude of gradient and MB-LBP are employed to extract further information from the original image, and two visual saliency methods are used to shrink the area that remains to be detected. In addition, a cautious combination algorithm has been designed to avoid the truncation and miss detection of pedestrians. The experiments demonstrate that the performance of our work is better than the models with only data-driven methods. The results show that the proposed feature expansion and background suppression can effectively extract the features and reduce the region of searching. The mAPs of the model have been improved by 2.98%, 8.52%, 6.18%, and 3.20%, respectively, on four datasets with different scenarios. In future work, we will adopt C++ to realize the algorithm and testing the time consumption. In addition, most of the parameters in this work have been optimized for this particular task. The more data collected in various scenarios require more optimizing investigation work that could focus on a type of model, in which the parameters could be self-adapted and trained.

## Figures and Tables

**Figure 1 sensors-20-05128-f001:**
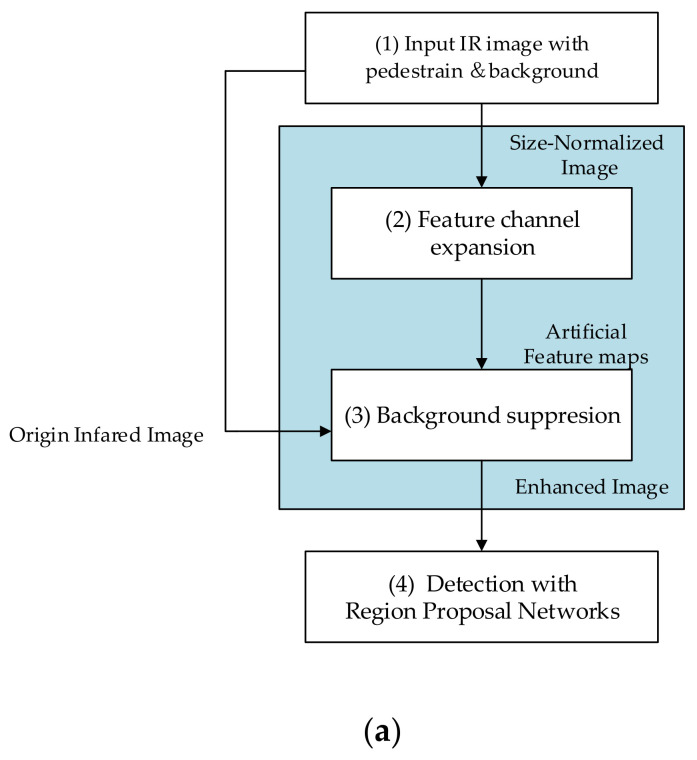
Introduction of the proposed approach: (**a**) Process of our model; (**b**) structure of the model.

**Figure 2 sensors-20-05128-f002:**
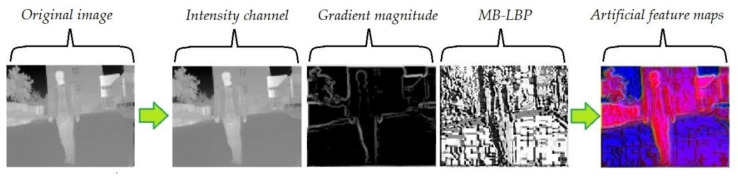
Channels of the proposed feature expansion with the algorithm: Intensity channel, gradient magnitude, multi-scale block local binary pattern (MB-LBP) texture channel.

**Figure 3 sensors-20-05128-f003:**
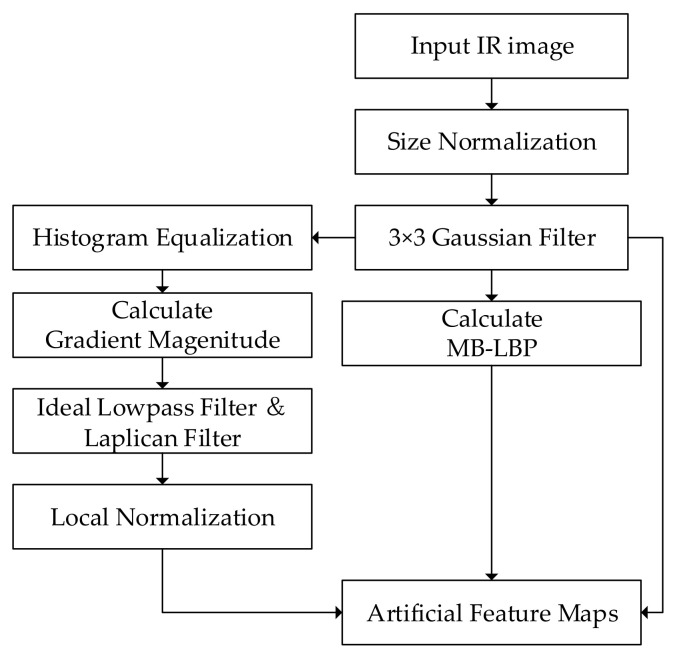
Flowchart of feature channel expansion.

**Figure 4 sensors-20-05128-f004:**
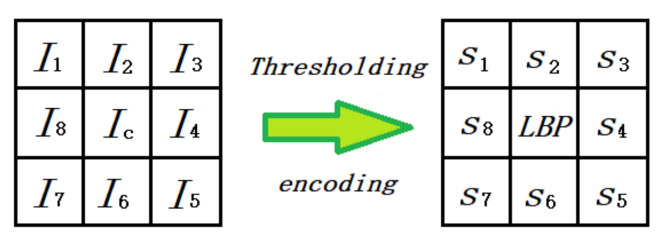
Basic LBP operator.

**Figure 5 sensors-20-05128-f005:**
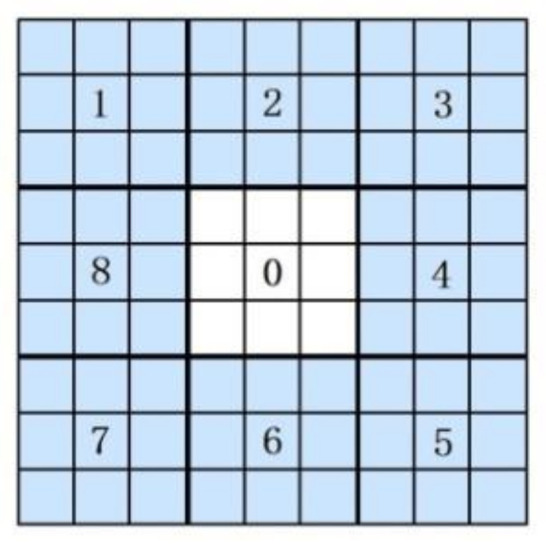
The 9 × 9 MB-LBP operator.

**Figure 6 sensors-20-05128-f006:**
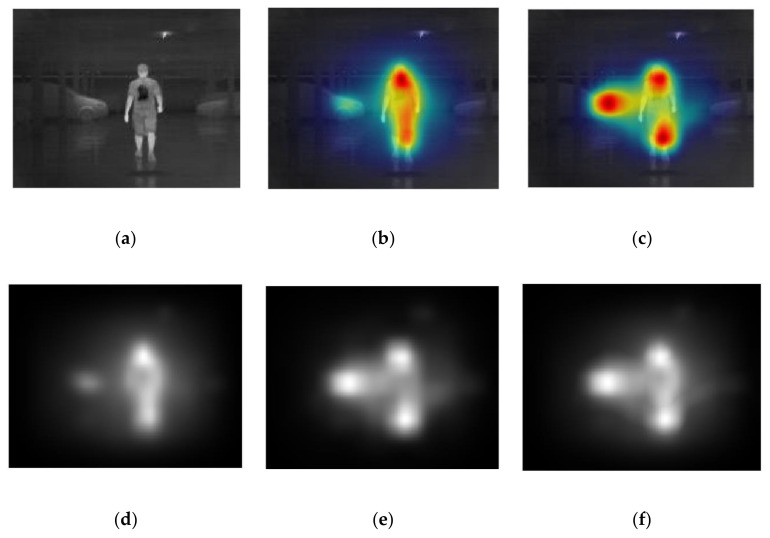
Images in the process of the saliency fusion: (**a**) The original image; (**b**) the heat map of GBVS saliency map; (**c**) the heat map of Itti-Koch saliency map; (**d**) intensity distribution of GBVS saliency maI(**e**) intensity distribution of Itti-Koch saliency map; (**f**) fusion of saliency map.

**Figure 7 sensors-20-05128-f007:**
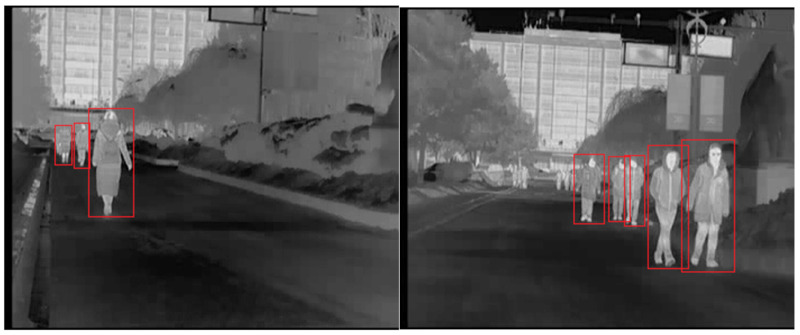
Some examples of CUSTFIR pedestrian dataset.

**Figure 8 sensors-20-05128-f008:**
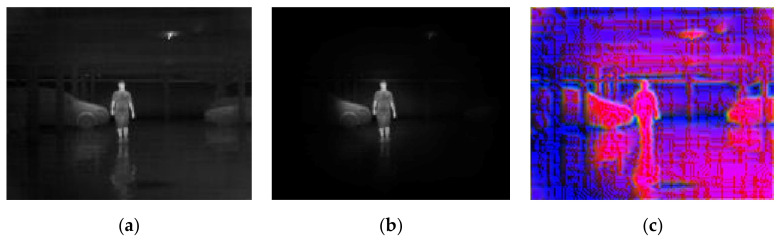
A simple example showing the model’s performance when it comes to the best conditions with (**a**) original infrared (IR) image, (**b**) background suppression result, and (**c**) channel expansion result.

**Figure 9 sensors-20-05128-f009:**
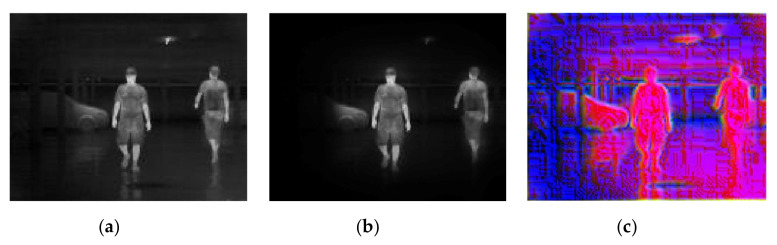
An example showing the model performance at the edge of the image with (**a**) original IR image, (**b**) background suppression result, and (**c**) channel expansion result.

**Figure 10 sensors-20-05128-f010:**
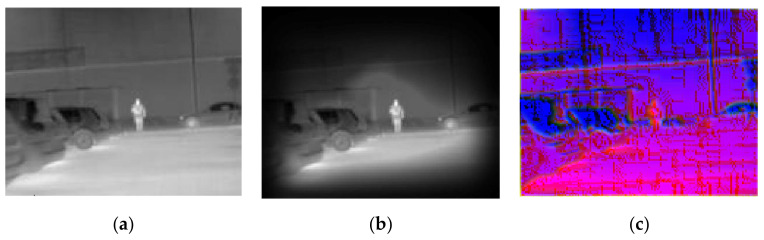
Example showing the model performance on a low contrast circumstance with (**a**) original IR image, (**b**) background suppression result, and (**c**) channel expansion result.

**Figure 11 sensors-20-05128-f011:**
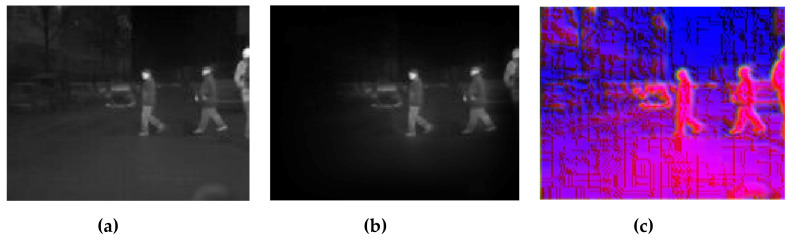
Example showing the model performance near the edge of the image with (**a**) original IR image, (**b**) background suppression result, and (**c**) channel expansion result.

**Figure 12 sensors-20-05128-f012:**
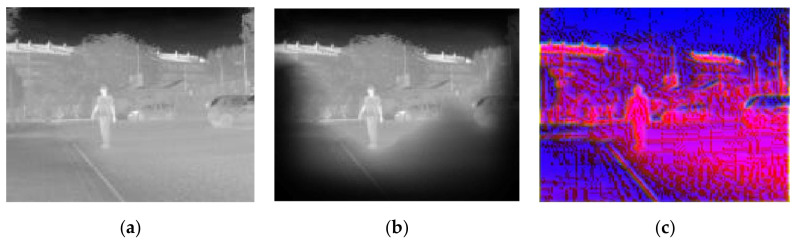
Example showing the model performance on a sunny day time outdoor with (**a**) original IR image, (**b**) background suppression result, and (**c**) channel expansion result.

**Figure 13 sensors-20-05128-f013:**
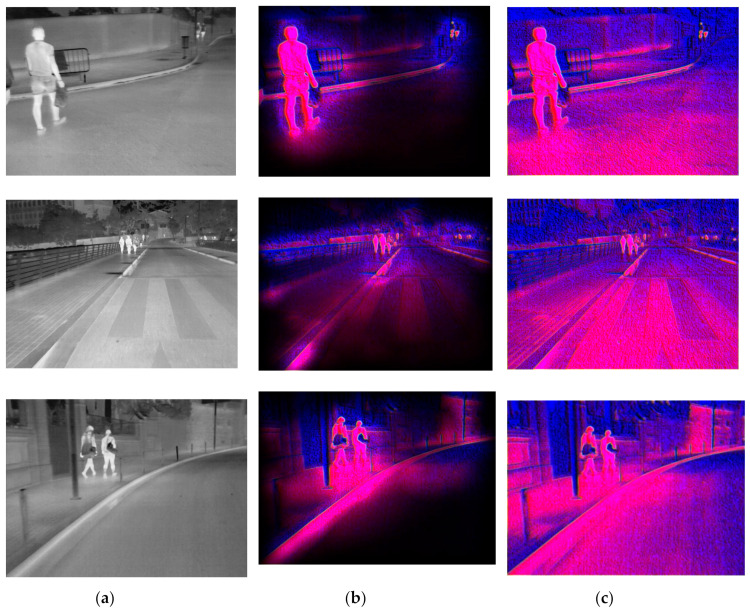
Examples of model performance using the CVC14 datasets with (**a**) original IR image (**b**) background suppression result on feature maps and (**c**) channel expansion result.

**Figure 14 sensors-20-05128-f014:**
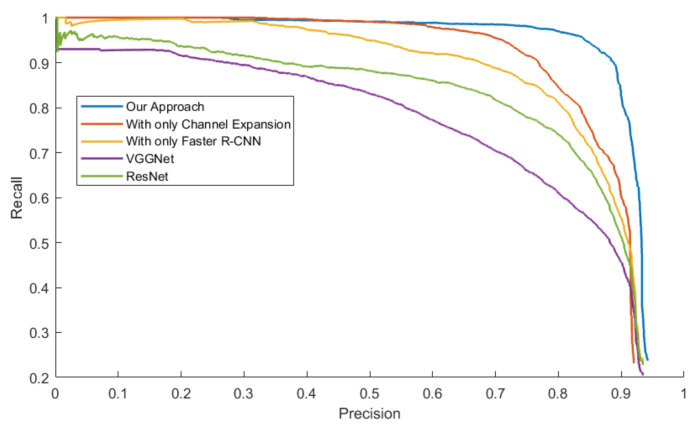
Precision–Recall (PR) Curve of pedestrian detection over LSI dataset by our approach (background suppression and channel expansion), the model with channel expansion only, and the model with the Faster R-CNN only, and other baselines.

**Figure 15 sensors-20-05128-f015:**
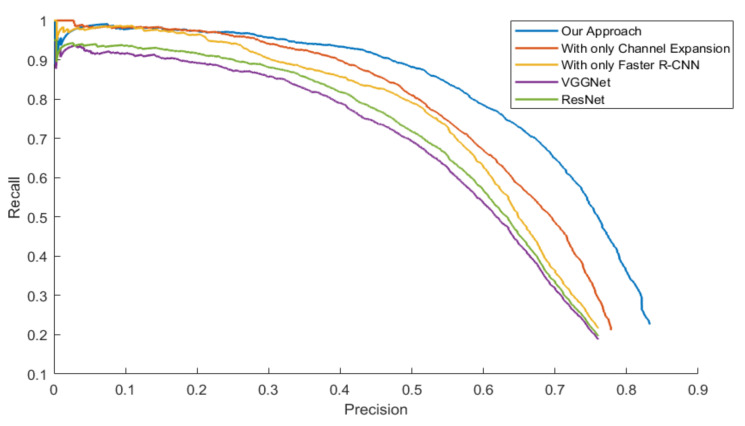
PR Curve of the pedestrian detection over CVC-14 Infrared pedestrian dataset by our approach, the model with channel expansion only, and the model with the Faster R-CNN only, and other baselines.

**Figure 16 sensors-20-05128-f016:**
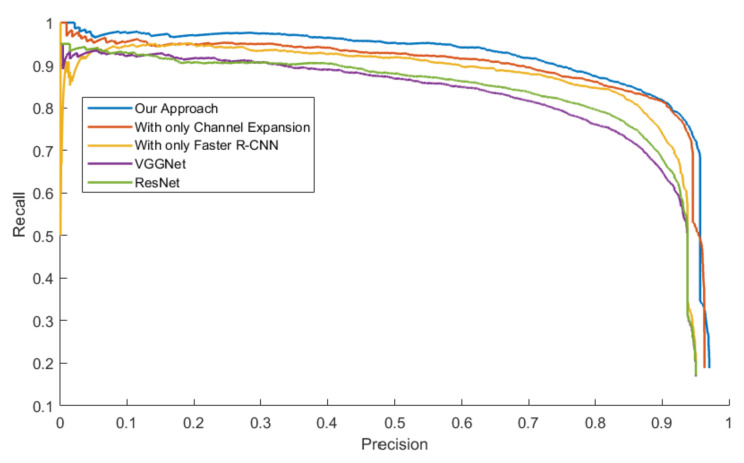
PR Curve of the pedestrian detection over CUST Infrared pedestrian dataset by our approach, the model with channel expansion only, and the model with the Faster R-CNN only, and other baselines.

**Figure 17 sensors-20-05128-f017:**
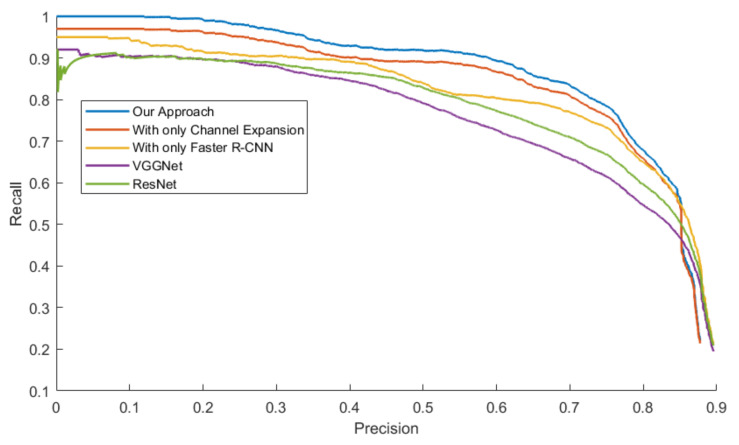
PR Curve of pedestrian detection over SCUT dataset by our approach (background suppression and channel expansion), the model with channel expansion only, and the model with the Faster R-CNN only, and other baselines.

**Figure 18 sensors-20-05128-f018:**
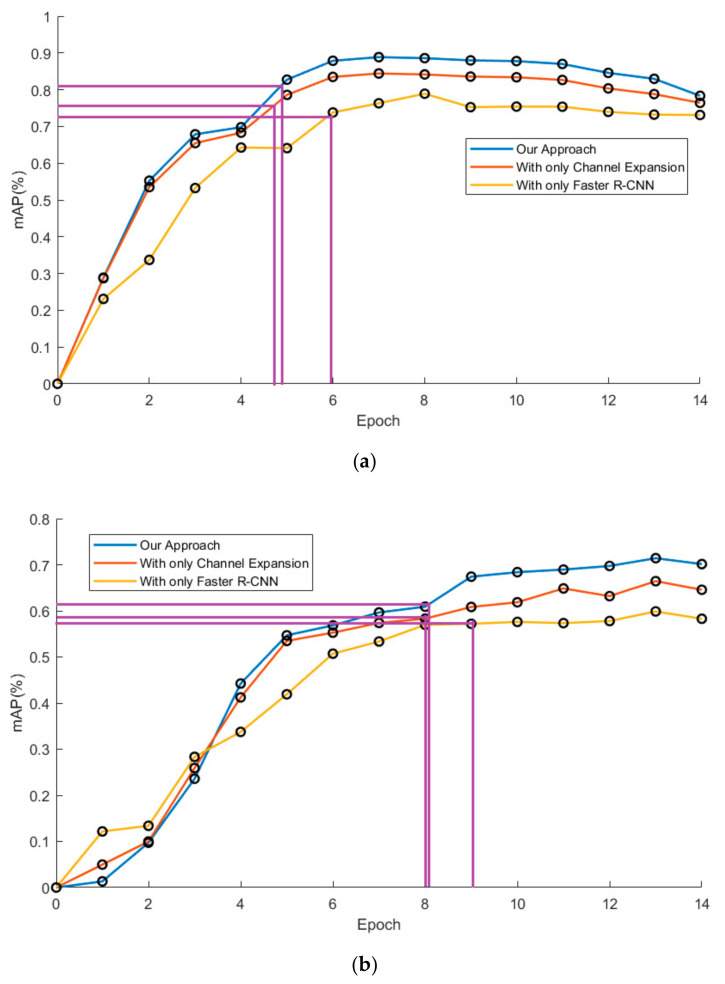
mAP versus epoch curve of pedestrian detection over LSIFIR dataset (**a**) and CVC-14 dataset (**b**) by our approach (background suppression and channel expansion), the model with channel expansion, and model with the Faster R-CNN.

**Table 1 sensors-20-05128-t001:** The architecture of feature optimized network in detail.

Layer	Number of Filters	Size of Feature Map	Size of Kernel	Number of Stride	Number of Padding
Convolution 1	64	960 × 640 × 64	3 × 3 × 64	1	1
ReLU 1		960 × 640 × 64			
Convolution 2	64	960 × 640 × 64	3 × 3 × 64	1	1
ReLU 2		960 × 640 × 64			
Max pooling 1	1	480 × 320 × 64	2 × 2	2	0
Convolution 3	128	480 × 320 × 128	3 × 3 × 128	1	1
ReLU 3		480 × 320 × 128			
Convolution 4	128	480 × 320 × 128	3 × 3 × 128	1	1
ReLU 4		480 × 320 × 128			
Max pooling 2	1	240 × 160 × 128	2 × 2	2	0
Convolution 5	256	240 × 160 × 256	3 × 3 × 256	1	1
ReLU 5		240 × 160 × 256			
Convolution 6	256	240 × 160 × 256	3 × 3 × 256	1	1
ReLU 6		240 × 160 × 256			
Convolution 7	256	240 × 160 × 256	3 × 3 × 256	1	1
ReLU 7		240×160×256			
Max pooling 3	1	240 × 160 × 256	2 × 2	2	0
Convolution 8	512	120 × 80 × 512	3 × 3 × 512	1	1
ReLU 8		120 × 80 × 512			
Convolution 9	512	120 × 80 × 512	3 × 3 × 512	1	1
ReLU 9		120 × 80 × 512			
Convolution 10	512	120 × 80 × 512	3 × 3 × 512	1	1
ReLU 10		120 × 80 × 512			
Max pooling 4	1	60 × 40 × 512	2 × 2	2	0
Convolution 11	512	60 × 40 × 512	3 × 3 × 512	1	1
ReLU 11		60 × 40 × 512			
Convolution 12	512	60 × 40 × 512	3 × 3 × 512	1	1
ReLU 12		60 × 40 × 512			
Convolution 13	512	60 × 40 × 512	3 × 3 × 512	1	1
ReLU 13		60 × 40 × 512			

**Table 2 sensors-20-05128-t002:** Information on infrared pedestrian datasets used.

Dataset	Size of Images	Number of Pedestrians	Number of Images	Sampling Frequency	Images Without Pedestrian
LSI	164 × 129	7624	6054	30 Hz	1424
CVC-14	640 × 471	8242	2614	30 Hz	632
SCUT	720 × 576	8679	4153	25 Hz	1322
CUSTFIR	352 × 288	6408	2836	25 Hz	621

**Table 3 sensors-20-05128-t003:** Events distribution of true positives, true negatives, false positives, false negatives.

		Prediction	
		Object	Background
Ground Truth	Object	True Positive	False Negative
	Background	False Positive	True Negative

**Table 4 sensors-20-05128-t004:** Quantitative comparison of the proposed approach, models only with channel expansion and only by deep learning approach (Faster R-CNN (convolutional neural networks) [40], ResNet [20], and VGGNet [19]) on LSIFIR, CVC14, SCUT, RIFIR datasets.

Metric (Datasets)	Our Work		Baselines		
	Channel Expansion and Background Suppression ^1^	With Channel Expansion Only	With Faster R-CNN Only [40]	ResNet with Sliding Window [20]	VGGNet with Sliding Window [19]
Precision (LSIFIR) ^2^	82.66% (81.62 ± 5.41%)	78.30%	75.39% (75.79 ± 3.65%)	75.70%	62.60%
Recall (LSIFIR)	82.15% (77.55 ± 3.23%)	78.36%	81.12% (79.97 ± 4.81%)	70.55%	70.45%
F-measure (LSIFIR)	82.40% (79.39 ± 2.17%)	78.33%	78.15 (77.78 ± 3.02%)	73.13%	66.29%
Precision (CVC-14)	67.41%	59.57%	56.55%	55.42%	52.90%
Recall (CVC-14)	69.70%	67.74%	69.23%	67.18%	63.95%
F-measure (CVC-14)	68.54%	63.39%	62.25%	60.73%	57.90%
Precision (SCUT)	66.73%	65.39%	62.17%	60.41%	59.57%
Recall (SCUT)	74.28%	71.86%	70.91%	69.70%	67.74%
F-measure (SCUT)	70.30%	68.47%	66.25%	64.72%	63.39%
Precision (CUSTFIR) ^3^	92.90%	92.93%	87.39%	84.78%	80.95%
Recall (CUSTFIR)	77.87%	76.98%	78.64%	77.90%	74.49%
F-measure (CUSTFIR)	84.72%	84.20%	82.79%	81.20%	77.58%

^1^ The approach proposed in this paper; ^2^ For the LSIFIR dataset, we ran our approach and Faster R-CNN five times and showed the mean and standard deviation as (mean ± std); ^3^ Datasets produced with our own IR camera.

**Table 5 sensors-20-05128-t005:** Comparison of Mean Average Precisions (mAPs) among our approach (enhancement with channel expansion and background suppression), part of our work (with channel expansion only), and the by deep learning approach only (Faster R-CNN [40], ResNet [20], and VGGNet [19]) on LSIFIR, CVC14, SCUT, CUSTFIR benchmark datasets.

Dataset	Our Work		Baselines		
	Channel Expansion and Background Suppression ^1^	With Channel Expansion Only	With Faster R-CNN Only [40]	ResNet with Sliding Window [20]	VGGNet with Sliding Window [19]
LSIFIR	82.72% (81.67 ± 1.54%)	80.80%	76.12% (78.69 ± 0.37%)	65.48%	60.23%
CVC-14	68.39%	61.86%	59.87%	57.34%	56.94%
SCUT	69.72%	66.81%	63.54%	59.72%	58.97%
CUSTFIR ^2^	85.45%	83.82%	82.25%	79.41%	78.10%

^1^ The approach proposed in this paper; ^2^ Datasets produced with our own IR camera.

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
