# Peer review of "Feature Channel Expansion and Background Suppression as the Enhancement for Infrared Pedestrian Detection"

_sensors, 2020, doi:10.3390/s20185128_

Round 1

Reviewer 1 Report

The manuscript looks at the detection of pedestrians from infrared imagery across a few different databases. This is a very interesting and important task, especially in light of advancements in autonomous vehicles. However, the approach used is not particularly innovative and the performance improvements for the additional processing are only minor.

General Comments

There are many instances throughout the manuscript where the authors state what they did without sufficient justification or background for why things were done that way or what the goal was. Additional justification for the approaches taken is required. For example, why are the image gradient magnitude and the orientation of each pixel in the input grey image calculated? Why and how were they combined with the intensity channel? What was the theory behind this approach and why was it selected over another? Since all of the features were already present in the original image why was it necessary to extract particular features and then reinsert them into different colour channels? Could a CNN have been trained to extract the same features if they were relevant?

Normalisation, especially histogram normalisation, has the potential to enhance noise. This noise enhancement is most prominent in channels with little or no salient information, such as when a target of interest is not present, in feature maps and/or when the input data contains artefacts. An example of this can be seen in Figure 2, where the image artefacts in the bottom right of the original image (caused by poor compression making the background grey levels blocky) are amplified in the MB-LBP feature map making it a very high contrast channel, particularly in regions of low salient information. This could also be the case in the gradient magnitude channel. If an image contained very little gradient information, such that the image was essentially devoid of interesting objects, applying histogram normalisation will simply amplify noise. Did the authors take this into account and did they have some limit to the normalisation process or was the CNN able to effectively ignore high contrast inputs that were essentially noise?

The use of existing saliency algorithms to de-emphasise regions of the image that are likely to be background is fine, but not really sufficient or significant enough to justify the manuscript. Many other people have used these techniques in many different areas. I would expect a larger contribution to the scientific literature than the use of a technique from the 1990’s combined with one from the mid 2000’s and then using a standard CNN for classification. The authors need to make a better case for why their approach is unique and worthy of publication. What is the new knowledge and approach shown in this manuscript?

How were the ground truths for the locations of the pedestrians determined, in particular for the CUSTFIR dataset?

The results figures (14, 15, 16 and 17) are called ROC curves but they are not. ROC curves have false positives on the x-axis and true positives on the y-axis. The authors are providing precision recall curves. This is fine, as they essentially show the same information, but they should not be identified as ROC curves.

Specific Comments

Line 22: "Four different parts of experiment are conducted..." – this phrase is not correct grammar and is not easily understood. After reading the manuscript I believe that the authors are referring to the four different datasets that they used. This needs to be clearly stated as it could also be understood to mean there were four difference experimental paradigms used to test the hypothesis from four separate angles.

Line 24: “The results imply…” – this is a scientific paper. It should not have to rely on implications or suppositions for its main finding. Either the results prove something, or they do not, they should not merely imply something. Please get more definitive scientific proof before making a claim and trying to publish your results.

Line 33-34: “…use of infrared imagery for pedestrian detection in visible images…” – infrared and visible images are different things. You can not use infrared imagery to detect things in visual images.

Line 34-35: “…particular interest, since they [infrared images] contain different information from the visible light images.” – having different information is not sufficient to say why it is helpful, you must say what different information is available and why that is useful for the task.

Line 42-43: “…[a] number of theories and methods have been put forward as solutions for visible light classification and detection problems…” – this statement is missing references. If a number of theories have been proposed you should reference some of them at least.

Line 47-48: “…visible light image with a size of 640x480x3 could record 921,600 pictures…” – I believe the authors have neglected the Bayer pattern present in most visual colour camera systems. This would result in only 307,200 pixels of unique information, which is exactly the same number as would be found in an infrared camera of the same resolution. As such the point the authors are making is invalid.

Line 49-51: “The IR cameras work in different wave ranges with various features, which means it is hard to find a unified approach for the many types of IR sensors.” – the fact there are differences does not automatically mean there are not some common approaches. The authors should outline some previous approaches and show that they are vastly different with nothing in common.

Line 60-61: “…some of them are designed to solve specific problems.” – this statement is missing references and details on some of the specific problems.

Line 70: “…deep CNN is still too expensive for real time applications.” – this is not necessarily the case. There are plenty of examples of real-time CNN deployments. I would give examples here, but Google Scholar gave me over 25,000 results when I searched ‘real-time deep CNN deployment’ so there are just too many to show. Additionally, most of the more recent results cover how to miniaturise and optimise for deployment on limited platforms (e.g. mobile phones) rather than how to perform the task.

Line 159-160: “Previous studies show that visual saliency methods are suitable for grayscale images, especially IR images without color distribution.” – this statement is missing a reference.

Line 187-188: “…the input image is first normalized…” – the authors should clearly describe exactly how normalisation of the images were achieved. Was it histogram normalisation (as indicated in line 207)? Why was one method selected over the other possibilities? Did normalisation achieve the goal of reducing the characteristics of different sensor devices? If so how do the authors know, and by how much did it reduce? If this normalisation was performed independently on each image were there flickering artefacts from changes occurring between images?

Figure 3: - if the input image is normalised why do each of the intensity channels need to be normalised? What would the result be if they were not normalised?

Line 281: “SCUT is a large far infrared pedestrian detection dataset.” – this database requires a reference.

Line 286: “The CVC14 day-night pedestrian sequence dataset…” – this database requires a reference.

Line 289: “CUSTFIR is a dataset obtained by our team…” – this database requires a reference. Is this database public like the others or is it private? If it is private it requires significantly more details to describe it since no one else can look at it.

Table 1: how many images were present in each dataset that did not contain a pedestrian? In all cases the number of pedestrians exceeded the number of images, which is not necessarily a realistic scenario. If all images contained at least one pedestrian then the system could use that to pick out the object in the scene that is most likely to be a pedestrian. It is an entirely different situation if pedestrians are rare or not in every image as then the system will have to determine if one is present at all.

Line 307: “A true positive is a case where the human area was correctly recognized as a human region.” – how did the CNN determine the location of a person? Was it via the generation of a bounding box? How was this matched with the annotated boxes from the datasets? How much overlap was required for a detection to be called a true positive?

Table 2 and Figure 7 both show the same basic concept. Neither is necessary, and there is certainly no need for both.

Line 321: “Accuracy is estimated as the average of best Precision and best Recall.” – if you did not directly measure accuracy, but rather estimated it from two other parameters you calculated, then why even report it as it is redundant?

Line 373: “The target is more distinct in the feature maps…” – please define and quantify what is meant by distinct. What metric is used to determine distinctiveness?

Figure 19: the claim in the text is “…that the mAP of our approach grows faster than that of the models with Faster R-CNN only.” However this is not necessarily the case. It is true that the final mAP value is higher in the authors approach than the Faster R-CNN only if this difference in maximum is accounted for it is not clear that the rate of rise of the system is any faster. In face it is the same or even slower in (b). It is also not clear that this is the case for all four data sets since only two are shown. The authors should define what they mean by the training time, for example the time to 90% of the final average mAP value, and then calculate the number of Epochs required to reach this point.

Reference List

There are so many problems and errors in the reference list that I will just recommend that the authors remove it and start again. I will list just a few of them below:

2. many details are missing, there are just generic placeholder names.

4. only 1 author is listed and then et al. This reference has four authors and 11. lists all four of that reference so there is no consistency.

8. is missing quotation marks around the title that all the others have.

11. is missing a publisher.

14. has 3 authors then et al. but other references list them all or only have one then et al.

15. has insufficient details.

26. is missing additional authors and conference details.

29. missing authors.

36. is a Wikipedia article cited in Feb 2017. Is this really the best reference to use for this piece of information?

Reviewer 2 Report

This paper proposes a novel channel expansion technique based on feature fusion is proposed to enhance the IR imagery and accelerate the training process. However, the paper is not innovative enough. Here are several issues below that should be well addressed.

  1. The authors needs to clearly indicate their contribution and the problems to be solved in Introduction.
  2. Fig.1 should show the structure of the model in detail, not just the process, so is Fig. 3.
  3. The experiments are not enough. The model should compare with other methods(baseline) in quantitative and qualitative analysis

4.Experimental analysis is not sufficient. Experimental analysis does not simply describe the increase or decrease of indicators. You should let readers know the nature of this problem.

5.In the “Introduction” or “Related work”, the recent works mentioned are not enough. Some representative methods related to saliency detection should be included in the paper, such as:

[1] Q. Zheng, SJ. Yu, XG. You. Coarse-to-fine salient object detection with low-rank matrix recovery. NEUROCOMPUTING, 376(2020), pp. 232-243. 

[2] J. Li, J. Yang, C. Gong, Q. LiuSaliency fusion via sparse and double low rank decomposition. Pattern Recognit. Lett., 107 (2017), pp. 114-122

[3] D. Zhang, J. Han, Y. Zhang, D. XuSynthesizing supervision for learning deep saliency network without human annotation. IEEE Trans. Pattern Anal. Mach. Intell. (2019)

Round 2

Reviewer 2 Report

The author has revised it according to the comments. However, there are a few minor flaws.

1.The author needs to describe the meaning of the marks in the figure, e.g. the yellow and red bounding boxes in Fig. 7.

2. The author needs to  point out the source of the baselines, though they are so popular.
